# Electrochemical Detection of Dopamine Based on Functionalized Electrodes

**Mathieu Ouellette** [1][ID]**, Jessy Mathault** [2]**, Shimwe Dominique Niyonambaza** [1,2]**, Amine Miled** [2][ID] **and Elodie Boisselier** [1,*][ID]

1   CUO-Recherche, Centre de recherche du CHU de Québec, Hôpital du Saint-Sacrement, Département d'ophtalmologie, Faculté de médecine, Université Laval, Québec, QC G3K 1A3, Canada

2   Electrical and Computer Engineering Department, Faculty of Sciences and Engineering, Université Laval, Québec, QC G1V 0A6, Canada

*   Correspondence: elodie.boisselier@fmed.ulaval.ca; Tel.: +1-418-682-7511 (ext. 84429)

**Abstract:** The rapid electrochemical identification and quantification of neurotransmitters being a challenge in the ever-growing field of neuroelectronics, we aimed to facilitate the electrochemical selective detection of dopamine by functionalizing commercially available electrodes through the deposition of a thin film containing pre-formed gold nanoparticles. The influence of different parameters and experimental conditions, such as buffer solution, fiber material, concentration, and cyclic voltammetry (CV) cycle number, were tested during neurotransmitter detection. In each case, without drastically changing the outcome of the functionalization process, the selectivity towards dopamine was improved. The detected oxidation current for dopamine was increased by 92%, while ascorbic acid and serotonin oxidation currents were lowered by 66% under the best conditions. Moreover, dopamine sensing was successfully achieved in tandem with home-made triple electrodes and an in-house built potentiostat at a high scan rate mode.

**Keywords:** cyclic voltammetry; neurotransmitters; electrochemical sensor; gold nanoparticles

---

## 1. Introduction

The World Health Organization (WHO) estimated in 2012 that 35.6 million people worldwide were suffering from neurological disorders which might be doubled by 2030 and more than tripled by 2050 due to the rapid aging of the population [1]. Yet, to this day, it has not been possible to establish a concrete and a solid link between ionic or molecular exchanges and brain electric signal propagation, and also degeneration and genesis of neurons in neurodegenerative diseases. The underlying cause is the complexity of neuronal interconnections and the difficulty to access deep brain zones. Among all the neurodegenerative diseases, Parkinson's disease is particularly well known for the large imbalance in neurotransmitter concentrations due to a dysfunction in dopaminergic neurons among other causes [2].

Several researchers have tried to address this problem by focusing on an implantable system. For example, Suzuki et al. proposed carbon nanotube multi-electrode array chips to selectively measure dopamine activity in brain slices [3]. They reported that they were able to sense a current of 22.6 $\mu$A/mm$^2$ for a dopamine concentration varying from 10 pM to 1 nM. In addition, Kimble et al. developed a real-time, wireless detection system to sense dopamine, adenosine, and glutamate during neurosurgery by using fast scan cyclic voltammetry (FSCV) [4]. However, this system was not designed for implantable devices that would enable monitoring the evolution of the disease in an active patient. Furthermore, Wang et al. reported a microsystem based on electroluminescence (ECL) to detect acetylcholine [5], that is, the detection of the light emitted by electroluminescent molecules during electrochemical reactions. The same technique has been also used to detect acetylcholine using

enzymatic biosensors based on cadmium sulfide nanocrystals [6]. Wang et al. also used ECL to detect 0.8 nM of choline and 1.7 nM of acetylcholine [5]. Additionally, Murari et al. proposed a new in vivo neurotransmitter detection method based on brain-implantable electrodes featuring an integrated potentiostat [7]. The proposed system is made of a sensor array with integrated microelectrodes and a pA potentiostat to detect the electrical activity of the brain. However, it does not formally identify the neurotransmitter generating the current. This is mostly due to the fact that the biointerface of the electrode is sensitive to its surrounding environment.

　　　Cyclic voltammetry (CV) is another technique used to detect neurotransmitters. It has been used, among other techniques, to detect dopamine, noradrenaline, serotonin, and indoleamine with carbon fiber electrodes because neurotransmitters oxidize at a low voltage [8,9]. It has been reported that FSCV was used in vitro, in vivo, on neuron cells and brain slices, and on anaesthetized or freely moving animals [10–12]. The voltammograms (current vs. voltage) obtained by these techniques are in theory specific to a given substance and may be used to determine precisely each neurotransmitter, as explained by Hermans et al. [13]. In electrochemistry, when voltage remains constant and only sensed current is varying, the technique is called amperometry. It offers a better temporal resolution than FSCV to detect catecholamines and indoleamine [14]. The temporal resolution of this technique is limited by two factors, namely, the diffusion of the neurotransmitter in the electrode and the electron transfer kinetic [13]. However, the reliability of these techniques remains limited in several applications.

　　　Despite CV being a well-known electrochemical analysis technique, the experimental conditions for electrochemical imaging of the brain are still in progress. Indeed, previous works were often limited to a single neurotransmitter in a highly controlled environment because of experimental conditions that would not allow selective detection of a specific neurotransmitter. Moreover, neurotransmitters such as dopamine and serotonin tend to oxidize rapidly (in less than 30 min) when exposed to ambient air and room temperature [15]. Additionally, some neurotransmitters are not electroactive [16]. Hence, research in this field faces several important problems where neurotransmitters degrade very rapidly, not all of them are electroactive, and their concentrations can be in the nanomolar range [17]. Separating different neurotransmitters is also known to be highly challenging and if it cannot be achieved, selectivity of the system should be exhaustively demonstrated. However, this is not always easily done or systematically reported in the literature [18,19]. Selectivity of dopamine depends on its buffer solution and on the quality of sensing electrodes. Other detection techniques, including optical-based detection techniques, have been developed as an alternative to electrochemical methods. However, detection of neurotransmitters remains challenging and no method surpasses the others. For example, the detection of dopamine and serotonin by surface-enhanced Raman spectroscopy (SERS) using a graphene–gold nanopyramid heterostructure provides an impressive sub-nanomolar detection limit [20]. Neurotransmitter-induced growth of gold nanoparticles as well as fluorescence-based detections have also been reported [21,22]. However, the equipment required for these techniques, including a bulky spectrometer for spectral recording, limits the in situ use, as well as the cytotoxicity of the nanomaterials involved, which is also a concern for in vivo detection.

　　　Several electrode functionalization methods have been proposed for the selective detection of dopamine (for a recent review, see reference [23]). Most of these techniques use carbon-based electrodes because of their advantages for biological applications [24]. For example, graphene electrodes functionalized with porphyrin [25], graphene oxide/multiwalled carbon nanotubes functionalized with hexadecyl trimethyl ammonium bromide [26], gold electrodes modified with polypyrrole-mesoporous silica molecular sieves [27], carbon quantum dot modified electrodes [6], reduced graphene oxide/polyethylenimine on a gold microelectrode [7], manganese tetraphenylporphyrin/reduced graphene oxide nanocomposite [8], or pretreated screen-printed carbon electrodes with NaOH [9] allow users to reach a limit of detection for dopamine from 1 μM to 8 nM (see Table S1, Supplementary Materials). Our system favors selectivity optimization, and functionalization with recognition molecules in conjunction with gold nanoparticles can be beneficial for designing multiplexed sensors.

In this work, we developed a biointerface that is more sensitive and selective for dopamine. The biointerface is functionalized with gold nanoparticles, DNA, and a polymer. Gold nanoparticles are known to have many advantages in electrochemistry in that they are highly conductive, allow a faster electron transfer kinetics, increase the electroactive surface of the electrode, and reduce the risks of overvoltage [28,29]. Moreover, depending on the synthesis method, they can be biocompatible and are consequently being used more and more frequently in nanomedicine [30,31]. Recent works have also highlighted the usefulness of gold nanoparticles in electrochemistry [32]. Reported nanoparticles, in that case, had a diameter of 60 nm and were deposed on the surface of the electrode during the electropolymerization. However, no pre-formed gold nanoparticles have previously been used. The size of the gold nanoparticles greatly influences their properties and therefore their efficiency in electrochemistry.

Several techniques based on nuclear medicine tomographic imaging, optical sensing, electrochemistry, or high-performance liquid chromatography are currently being developed to reach the limit of detection in the range of nM, even pM and fM [33–36]. Despite this extreme sensitivity, the selectivity remains an issue. This manuscript proposes a new strategy to improve probe selectivity in order to target in vivo experiments. A new functionalization protocol based on gold nanoparticles that were already reduced by a modified Brust method has been developed [37] in order to optimize their properties and thus the detection of neurotransmitters. In order to increase the sensitivity of the probe for a given neurotransmitter, the biosensor is also functionalized with DNA and a monomer (*o*-phenylenediamine (*o*-PD)). The principle of this functionalization relies on the possibility for DNA to bind specifically (and with a high affinity) to neurotransmitters via hydrogen bonds and electrostatic and intercalation interactions. Gold nanoparticles are used to increase the sensitivity of the electrode and amplify its electrochemical response. Upon electropolymerization, the monomers polymerize and entrap both the DNA and the targeted neurotransmitter at the surface of the electrode, conferring durability and stability to the model. A washing is then performed to remove just the neurotransmitter used as the template. The binding sites remain unchanged and the given neurotransmitter may again bind to the DNA during its electrochemical dosage. This protocol has already been validated for dopamine with a commercial electrode [38]. However, it had not yet been tested on smaller fiber electrodes. In this paper, we thoroughly investigated the influence of different functionalization conditions. Different potentiostats (in-house built and commercial) and electrodes (custom-made and commercial) were used to detect dopamine with an enhanced selectivity.

## 2. Materials and Methods

Dopamine hydrochloride, serotonin hydrochloride, Tris base, sodium acetate, DNA sodium salt from salmon testes (#D1626), and polyethylene glycol methyl ether thiol with a molecular weight of 800 g·mol$^{-1}$ (referred to as PEG$_{800}$) were purchased from Sigma-Aldrich (St Louis, MO, USA). Dichloromethane and methanol were purchased from Thermo Fisher Scientific (Hudson, NH, USA). Ascorbic acid, *o*-phenylenediamine (*o*-PD), acetic acid, sodium chloride (NaCl), potassium chloride (KCl), sulfuric acid (H$_2$SO$_4$), disodium hydrogen phosphate (Na$_2$HPO$_4$), monobasic potassium phosphate (KH$_2$PO$_4$), gold chloride trihydrate (HAuCl$_4$·3H$_2$O), sodium borohydride (NaBH$_4$), chlorhydric acid (HCl), and nitric acid (HNO$_3$) were purchased from VWR International (Monroeville, PA, USA). Ethanol was obtained from Greenfield Global (Richmond Hill, ON, Canada).

### 2.1. Potentiostats

A compact potentiostat (referred to as in-house built potentiostat) was designed, as shown in Figure 1 (see Figure S1, Supplementary Materials, for the schematic of the experimental setup).

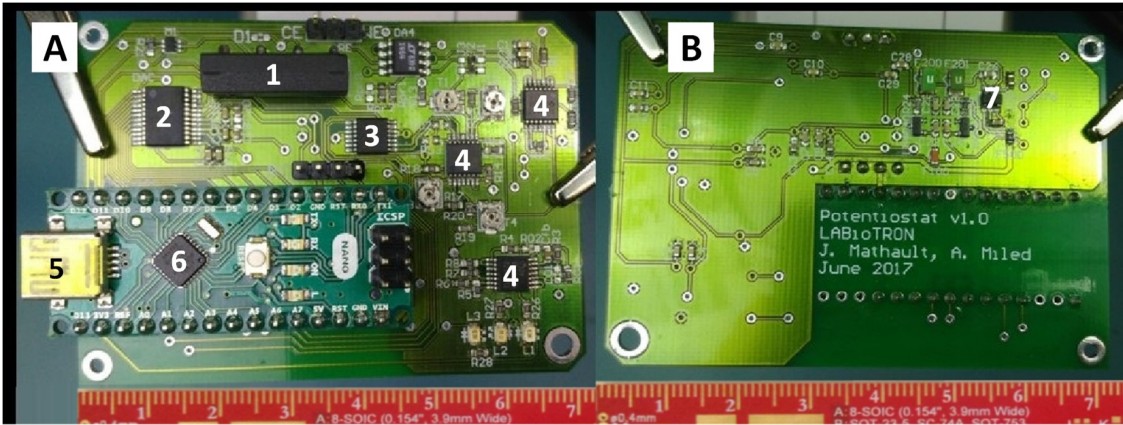

**Figure 1.** (**A**) Top view and (**B**) back view of the compact in-house built potentiostat. 1, power relay; 2, digital-to-analog converter; 3, digital potentiometer; 4, operational amplifier; 5, universal serial bus connector; 6, ATmega328 microcontroller; 7, LM2776 voltage converter.

The designed potentiostat is a one-channel biosensor that can handle only one measurement at a time. Stimulation voltage waveforms generated by the potentiostat can reach 100 Hz, while amplitudes vary between −2.5 and 2.5 V. The negative voltage is generated by the circuit using a charge-pump voltage converter (LM2776) from Texas Instruments Inc. (Sherman, TX, USA). The sensed current range varies between ±100 nA, ±500 nA, ±1 µA, ±5 µA, ±50 µA, ±100 µA, ±500 µA, ±1 mA, or ±3 mA. The minimum current resolution is 0.2 nA.

The potentiostat is composed of a digital data processing unit and an analog part. The data processing part generates all digital stimulations through an ATmega328P microcontroller (Microchip Technology, Chandler, AZ, USA). The potentiostat is connected to a computer through a USB interface. The user can set the scan rate, minimum and maximum voltages, and current range through a home-designed interface on LabVIEW (Version 2018, National Instruments, Austin, TX, USA). The digital part of the potentiostat also handles the feedback loop in order to stabilize the stimulation voltage. A detailed description of the circuit used has been previously reported [39]. Current range was set through programmable resistors and the designed software interface. The potentiostat is powered directly by the USB cable (5 V). Furthermore, the same sensor can also handle wireless communication with the interface through a Bluetooth module. Results obtained from the designed potentiostat were compared with a commercial one and both were consistent. This electrochemical biosensor is designed to work in the three-electrode configuration (working, counter, and reference electrodes). Along with the in-house built potentiostat, the WaveNow™ potentiostat from Pine Research Instrumentation Inc. (Durham, NC, USA) was also used, referred to as commercial potentiostat with its operating software AfterMath™ (version 1.3.7259).

Despite the fact that differential pulse voltammetry (DPV) could provide better performance, this paper focuses on the CV technique. DPV is expected to improve sensitivity due to reduced background effects. However, our main concern in this paper was the functionality. Both CV and DPV should be effective when the concentration is high, which explains our intentional choice of high concentrations in this work. Furthermore, in this paper, we focused on electrode functionalization, and the potentiostat and functionalized electrodes were considered as a unique system.

The proposed potentiostat is actually designed to be a future embedded system for cell culture, as part of a complex system which will include several other modules (impedance and capacitive sensors). Different architectures of potentiostats are reported in the literature, but in this paper, we considered both the potentiostat and electrode functionalization as a complete system. Indeed, the potentiostat would be useless without coupling it with a selectivity strategy. Thus, the novelty of this article lies in the presented electrochemical sensor, considering both together, that is, the electronic sensor with functionalized electrodes as a detection device.

## 2.2. Custom-Made Electrodes

Four different triple electrodes were designed by Doric Lenses Inc. (Quebec City, QC, Canada) as follows (see the dimensions and the configuration in Figure S2, Supplementary Materials). The dimensions of the triple electrodes and their setup were chosen according to their potential use of mouse brain slices. These electrodes could thus be used for in vitro experiments for further experiments. One probe, referred to as tungsten 125 triple electrode, had 2 electrodes of tungsten (W) with a 125 μm diameter and 1 electrode of platinum–iridium (Pt/Ir) alloy with a 100 μm diameter. One electrode, referred to as tungsten 75 triple electrode, had 2 electrodes of tungsten (W) with a 75 μm diameter and 1 electrode of platinum–iridium (Pt/Ir) alloy with a 100 μm diameter. One probe, referred to as carbon triple electrode, had 2 electrodes of carbon (C) with a 30 μm diameter and 1 electrode of platinum–iridium (Pt/Ir) alloy with a 100 μm diameter. The last one, referred to as stainless triple electrode, had 2 electrodes of stainless steel with a 50 μm diameter and 1 electrode of platinum–iridium (Pt/Ir) alloy with a 100 μm diameter. The tip length of all the electrodes was 200 μm. Scanning electron microscopy (SEM) images were captured using a Quanta™ 3D FEG scanning electron microscope from FEI Company (Hillsboro, OR, USA) (Figure S3, Supplementary Materials). The images confirmed the expected diameters of fibers for each electrode. This analysis was performed after several electrochemical experiments, explaining some observable deposits on the electrodes.

Doric Lenses Inc. also produced carbon fiber electrodes with a 30 μm diameter and a 200 μm long tip. Fiber working and counter electrodes and a commercial Ag/AgCl reference electrode from Pine Research Instrumentation Inc. (#RRPEAGCL) were mounted on a 3D-printed frame to create the so-called three-electrode setup. The RRPE1002C screen-printed carbon electrodes from Pine Research Instrumentation Inc., with a 4 mm × 5 mm carbon working electrode, a carbon counter electrode and a Ag/AgCl reference electrode, are referred to as commercial electrodes.

A correlation exists between brain damage, such as penetration trauma, and the size of an implantable device [40]. Even if smaller electrodes up to 7 μm induce a much weaker immune response than 200 μm microdialysis probes, their surface is very limited for the detection of neurotransmitters [41,42]. Thus, compared to the existing ones, the electrodes used in this study would be adequate for in vivo data collection, inducing nonetheless minor to moderate tissue damage.

## 2.3. Synthesis of Gold Nanoparticles

The functionalization protocol was based on the use of a bioimprinted polymer previously described in the literature [32]. However, in our case, instead of producing gold nanoparticles in situ during the functionalization process, we attempted to use gold nanoparticles that were already reduced and stabilized before the functionalization.

All the glassware used for the synthesis of gold nanoparticles was first thoroughly washed with aqua regia (3:1 $HCl:HNO_3$) and rinsed with Nanopure water. The synthesis was performed using a modified method of de Oliveira et al. [43]. Briefly, a 0.5 mL amount of a 0.1 g/mL stock solution of $HAuCl_4 \cdot 3H_2O$ was first diluted in 30 mL of water:methanol 1:1 (*v/v*). Then, 0.085 g of $PEG_{800}$-SH and 10 mL of methanol were added to the flask and the solution was left to agitate for 15 min. Subsequently, a 0.050 g amount of $NaBH_4$ was dissolved in 20 mL of ice-cold Nanopure water, and kept in an ice bath while it was added dropwise to the gold solution using a peristaltic pump (VWR® Variable-Speed Peristaltic Pump, VWR International, Monroeville, PA, USA) (1 mL/min) under vigorous stirring (600 rpm). The mixture was then left to agitate for three more hours to ensure proper stabilization of the gold nanoparticles [44]. The methanol was then removed using a rotary evaporator (Rotavapor-R, Buchi, New Castle, DE, USA) and the gold nanoparticles were extracted using dichloromethane and a minimal amount of brine. The dichloromethane was then dried using a rotary evaporator and the nanoparticles were redissolved in 10 mL of Nanopure water. They were then dialyzed three times against Nanopure water over 24 h to remove any remaining reagent. The final concentration of gold nanoparticles was determined after precisely weighing a freeze-dried portion of the solution. The gold nanoparticles were then characterized using ultraviolet–visible (UV-vis) spectroscopy, transmission electron microscopy (TEM) and dynamic light scattering (DLS).

UV-vis spectra were recorded using the Varian Cary 50 Bio UV-vis, from Agilent Technologies (Santa Clara, CA, USA), and disposable plastic cuvettes (#759200, 10 mm × 10 mm pathlength) from Brand (Wertheim, Germany) were used.

For TEM analysis, copper grids covered with a vaporized carbon film were purchased from Ted Pella (Redding, CA, USA). Gold nanoparticles were deposited on the grid and left for one minute before removing the excess volume with a filter paper. The image acquisition was performed with a JEM 1230 microscope from JEOL (Tokyo, Japan). The zoom factor was 80,000×. More than 500 gold nanoparticles were counted and analyzed with ImageJ (Version 2016, National Institutes of Health, Bethesda, MD, USA) to determine the number and the diameter of the gold cores.

DLS was performed with the NanoBrook Omni analyzer from Brookhaven Instruments Corporation (Holtsville, NY, USA). The solutions were analyzed in 10 mM $KNO_3$ at 25 °C at an angle of 90° using the CONTIN mode. After an equilibrium time of 10 min, 10 measurements of 120 s each were performed and the effective diameter obtained is reported in this study.

*2.4. Functionalization of the Electrodes*

2.4.1. Stock Solutions

Stock solutions were prepared so that they could be added in a small quantity to another solution in order to reach a given final concentration. A 625 mg/L DNA stock solution was prepared in a 0.1 M Tris buffer (pH 7.0). It was then separated in aliquots that were stored at −20 °C. Three different stock solutions (10 mM dopamine, 10 mM ascorbic acid, and 10 mM *o*-phenylenediamine (*o*-PD)) were freshly prepared daily in either pH 7.0 PBS 10× (primary conditions used unless otherwise specified) or Nanopure water (optimized conditions), depending on the experiment, and then kept at 4 °C protected from light as much as possible. Other solutions used were a concentrated acetate buffer (2.5 M, pH 4.8) containing NaCl (0.1 M), an electrode pretreatment solution (0.05 M $H_2SO_4$, as proposed by Pine Research Instrumentation Inc. for these electrodes [45]), and a washing solution (ethanol:acetic acid 80:20 *v/v*).

2.4.2. Pretreatment Conditions

In order to activate the surface of the carbon commercial electrode, a single CV cycle was performed from −2.5 to 2.5 V with the commercial potentiostat with a scan rate of 100 mV/s in the pretreatment solution [45].

2.4.3. Functionalization Solution

In order to functionalize the commercial electrodes for dopamine, a 10 mL portion of a functionalization solution was prepared as follows (but different parameters were tested, as detailed in the Results and Discussion section). First, 1 mL of the dopamine and 40 µL of the DNA stock solutions were added to 2 mL of the acetate buffer and agitated for 5 min. The gold nanoparticle solution was then added to a final concentration of 0.1 mg/mL, followed by 1 mL of the *o*-PD solution. The volume was adjusted to 10 mL by adding Nanopure water. The functionalization solution was agitated and, depending on the experiment, sonicated (primary conditions) or not (optimized conditions). The solution then had to be used immediately for the functionalization and could not be used a second time because its constituents were consumed in unknown proportions. In the case where the pre-formed gold nanoparticles were not used, $HAuCl_4 \cdot 3H_2O$ 0.01% *m/v* was instead added to the solution (10 µL of an aqueous 0.1 g/mL solution) as previously reported [32].

2.4.4. Functionalization Conditions

Commercial electrodes were immersed in the functionalization solution for 5 min prior to starting the electropolymerization by CV using the commercial potentiostat. Unless otherwise specified (see Results and Discussion section), five cycles were then performed from −0.2 to 0.9 V at a scan rate of

50 mV/s. Once completed, the electrode was gently removed from the solution and left to dry for 1 h before any washing procedure.

### 2.4.5. Washing and Storage

The electrode was finally washed twice during 15 min either in the washing solution (primary conditions) or in PBS (optimized conditions), depending on the experiment (see Results and Discussion section). It was then thoroughly rinsed with Nanopure water. Once dried, the functionalized electrode was kept at 4 °C and protected from dust.

### 2.5. Analysis of Neurotransmitters

Using the commercial potentiostat, two cycles of CV were performed from −0.3 to 0.9 V with a scan rate of 50 mV/s. Only the second cycles of the voltammograms were considered because of the instability of the first cycle due to electrode activation. A PBS blank was always analyzed first to ensure electrode integrity, followed by the analyte and PBS again (the latter being subtracted from the analyte).

When the in-house built potentiostat was used, CV frequency, sampling time, and CV duration had to be tuned in order to obtain the optimal response. Typical values were, respectively, 5 Hz, 0.001 s, and 1 s. The maximum sensed current varied between 0.1 and 10 μA. The stimulation was always triangular, and the experiments were performed between −1 and 1 V.

## 3. Results and Discussion

### 3.1. Synthesis and Characterization of the Gold Nanoparticles

Following the successful gold nanoparticle synthesis and purification (see Section 2.3 in Materials and Methods), the gold nanoparticles were analyzed by three different techniques. (1) UV-vis spectroscopy shows a characteristic plasmon band located at 512 nm (Figure S4, Supplementary Materials), (2) TEM imaging reveals an average core size diameter of 2 ± 1 nm (Figure S5, Supplementary Materials), and (3) DLS indicates that the mean hydrodynamic diameter of the nanoparticles is 16 ± 1 nm. DLS measurements take into account the gold core, the stabilizing ligands, and the hydration sphere. Therefore, our pre-formed gold nanoparticles are coated by a polymeric ligand but are still significantly smaller than the ones used previously in the literature (bare gold nanoparticles of 60 nm formed in situ) [32].

### 3.2. Influence of the Software Parameters on the Detection of Dopamine

The influence of different parameters on the dopamine detection was investigated. Figure S6 (Supplementary Materials) shows the effect of tuning the maximum sensed current from 0.1 to 50 μA on the detection of 10 mM dopamine while using the in-house built potentiostat and the carbon triple electrode (see Section 2.2, pictured in Figure 2B). Below 0.5 μA, it is not possible to detect the entire signal, whereas at 10 μA and above, the signal becomes noisy. Between these values, the voltammograms are clear and show defined oxidation and reduction peaks at amplitudes between −0.5 and 0.5 μA. Figure 2A presents the result obtained with the commercial potentiostat for the same concentration of dopamine and electrode.

$$\text{Sampling rate} \ = \ 2 \times (V_{\text{max}} - V_{\text{min}}) \times \text{Frequency} \tag{1}$$

According to Equation (1), where $V_{\text{max}}$ and $V_{\text{min}}$ are, respectively, the maximum and minimum voltages during the CV experiment, a frequency of 5 Hz used with the in-house built potentiostat with a voltage scan from −0.8 to 1 V is equivalent to a scan rate of 18 V/s. We could not reach this value with the commercial potentiostat since it is limited to 10 V/s, but at 5 V/s, the general shape of the voltammogram is greatly affected when using the triple electrode (dotted line in Figure 2A). A clear oxidation peak is not detectable, while the reduction peak is decreased to 0.15 μA only. At a lower scan rate (50 mV/s), even the reduction peak is lost (solid line in Figure 2A). When using the three-electrode setup (see

details in Section 2.2 in Materials and Methods), it was never possible to obtain a good measurement and only noise was sensed (Figure 2C, data not shown). In this case, the three-electrode setup may not allow the electrodes to be sufficiently close to each other and guarantee dopamine detection.

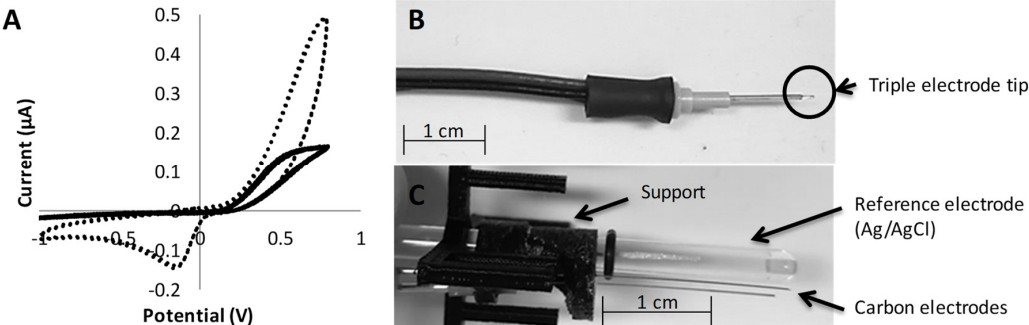

**Figure 2.** (**A**) Voltammogram obtained for 10 mM dopamine using the carbon triple electrode with the commercial potentiostat at a scan rate of (solid line) 50 mV/s and (dotted line) 5 V/s. (**B**) Picture of the carbon triple electrode used for the measurements. (**C**) Picture of the three-electrode setup.

### 3.3. Study of the Sensitivity of Dopamine Using Triple Electrodes Based on Different Materials

The limit of sensitivity of four different triple electrodes was investigated. Figure 3 shows the voltammograms of three concentrations of dopamine (10, 1, and 0.1 mM) obtained after the subtraction of the PBS signal while using the in-house built potentiostat and stainless, tungsten 75, tungsten 125, and carbon triple electrodes. For the stainless (Figure 3A) and tungsten 75 (Figure 3B) triple electrodes, dopamine could not be detected even at a concentration of 10 mM (in blue). For tungsten 125 (Figure 3C), 10 mM dopamine induces a very strong response (350–400 µA), however the electrode does not detect 1 mM of the same analyte (in red). With the carbon triple electrode (Figure 3D), the detection of dopamine at 1 mM is possible (amplitude of 0.4 µA), although 0.1 mM (in green) is not detectable. Consequently, carbon electrodes were chosen for the subsequent experiments because this material offered the best sensitivity towards dopamine.

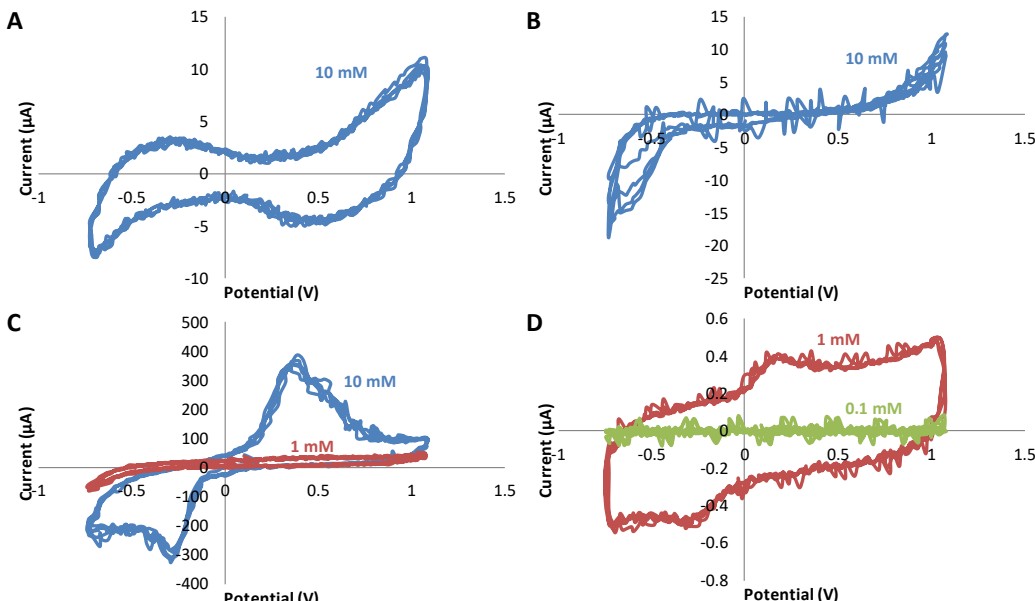

**Figure 3.** Determination of the sensitivity towards dopamine for different triple electrodes: (**A**) stainless, (**B**) tungsten 75, (**C**) tungsten 125, and (**D**) carbon. Three concentrations were used: 10 mM (in blue), 1 mM (in red), and 0.1 mM (in green).

Regarding possible in vitro experiments, it has been reported that the functionalization of electrodes can highly improve their sensitivity and selectivity towards a specific neurotransmitter [32], and this approach is of outmost interest. We thus investigated the influence of different functionalized electrodes on the sensitivity and the selectivity of dopamine detection.

### 3.4. Effect of Different Functionalization Conditions on the Sensitivity and Selectivity of Dopamine Detection

In light of the results previously obtained (Figures 2 and 3), the in-house built potentiostat was suitable based on its sensitivity for dopamine when used in tandem with the triple electrodes. A high scan rate is important to detect dopamine through both a rapid oxidation and reduction. However, in order to minimize the cost of high-throughput functionalization experiments, commercial electrodes and potentiostat were used. The detection of dopamine was thus achieved with commercial electrodes at the scan rate of 50 mV/s (Figures 4 and 5). Indeed, while as previously mentioned this slow scan rate is not suitable with the home-made glass fiber triple electrodes (Figure 2B), it produces good results with the commercial electrodes.

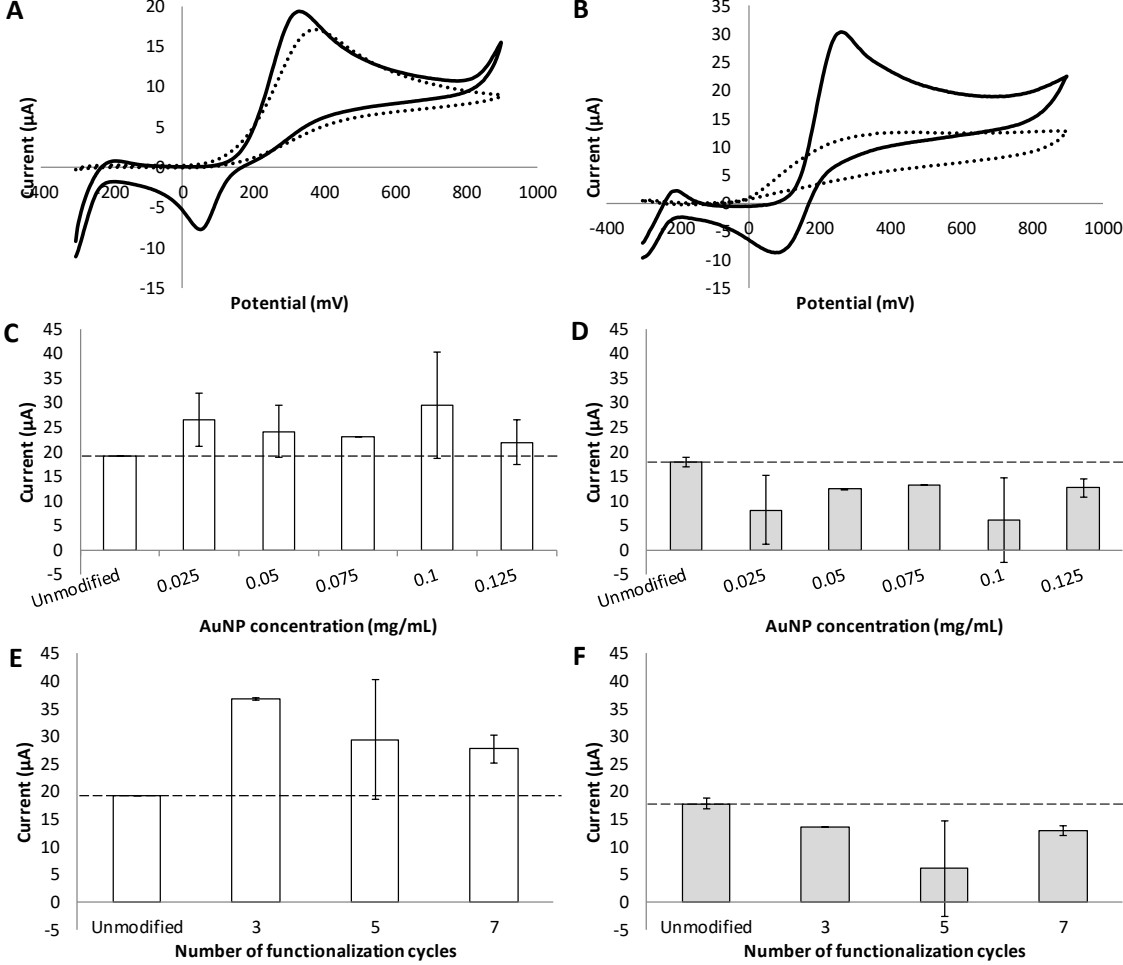

**Figure 4.** Voltammograms obtained for the analysis of (solid black) 1 mM dopamine and (dotted) 1 mM ascorbic acid for (**A**) an unmodified commercial electrode and (**B**) a commercial electrode functionalized as previously described with our pre-formed gold nanoparticles (here with 0.05 mg/mL of gold nanoparticles using 5 CV cycles). The currents obtained for the functionalized electrodes (**C,D**) with different concentrations of gold nanoparticles in the functionalized solution or (**E,F**) with different numbers of cycles for the functionalization are depicted in histograms (dopamine in white bars and ascorbic acid in grey bars); in these cases, the other parameters were fixed.

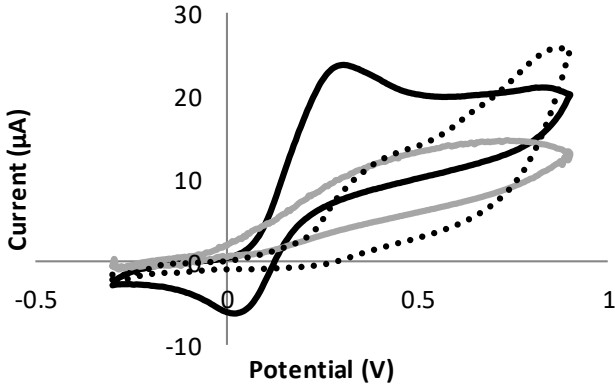

**Figure 5.** Analysis of (full black line) 1 mM dopamine, (full grey line) 1 mM ascorbic acid, and (dotted black line) 1 mM serotonin with commercial electrodes that have been functionalized for a higher selectivity towards dopamine.

The detection of dopamine was thus performed with commercial electrodes without (see Figure 4A) and with (see Figure 4B) the functionalization process. Without functionalization, both 1 mM dopamine and 1 mM ascorbic acid are detected and produce a current of 15–20 µA (Figure 4A). When the electrodes are functionalized using gold nanoparticles, DNA, and polymers (see Section 2.4 in Materials and Methods), the dopamine peak always remains sharp and its amplitude increases (peak at 30 µA), whereas the ascorbic acid peak becomes poorly defined and with a lower current amplitude (10–15 µA; Figure 4B). It should be noted that the small peak observed sporadically on the voltammogram at around −200 mV is an artifact due to the subtraction of the PBS signal from the CV. The layer added on the electrodes through the functionalization process allows better sensitivity and selectivity for the detection of dopamine. The influence of different parameters involved in this process was then investigated.

The concentration of gold nanoparticles used in the functionalization solution (Figure 4C,D, respectively for dopamine and ascorbic acid) as well as the number of cycles used for the electrodeposition of the film (Figure 4E,F) have no significant influence on the subsequent analysis of 1 mM dopamine or 1 mM ascorbic acid. It is important to mention that it was not always clear whether a real peak was visible on a voltammogram. For example, some poorly defined peaks, usually at low amplitudes, had current values in the order of 10 µA, while others were simply not considered and were assigned the value of 0 µA. A standardized threshold should be established to determine whether a voltammogram shows a real peak. Nonetheless, the general trend is that as long as the electrode is functionalized, the amplitude of the dopamine oxidation peak increases compared to the unmodified electrode and the ascorbic acid peak decreases. For dopamine, the unmodified electrode has a current amplitude of 19 µA (symbolized by the dotted line in Figure 4C), and the values reach 26, 24, 23, 29, and 22 µA when electrodes are functionalized with solutions containing 0.025, 0.05, 0.075, 0.1, and 0.125 mg/mL of gold nanoparticles, respectively (Figure 4C). For ascorbic acid, at the same gold nanoparticles concentrations, the current amplitudes are 8, 12, 13, 6, and 13 µA, while the unmodified electrode has a current amplitude of 18 µA (symbolized by the dotted line in Figure 4D). For 3, 5, and 7 functionalization cycles, current amplitudes of 37, 29, and 28 µA are observed for dopamine (Figure 4E) and 14, 6, and 13 µA for ascorbic acid (Figure 4F). Furthermore, using HAuCl$_4$ in the functionalization solution with optimized conditions (see Section 2.4 in Materials and Methods) leads to results similar to those obtained with pre-formed gold nanoparticles (Figure S7, Supplementary Materials). Despite the fact that the concentration of gold nanoparticles as well as the number of cycles during the functionalizing process do not drastically influence the dopamine detection, functionalized electrodes allow a better sensitivity and selectivity. Thereafter, it could be interesting to assess the response of these functionalized electrodes in the presence of different mixtures of molecules in several ratios.

Figure 5 compares the voltammograms of 1 mM of three different neurotransmitters following the functionalization of commercial electrodes in order to make them more selective to dopamine. All the experiments presented in Figure 5 were performed using the optimized functionalization parameters, as in Figure S7, Supplementary Materials (less salts in the functionalization solution, no sonication, and washing in "soft" conditions, see Section 2.4 in Materials and Methods). Indeed, using less salt (Na$^+$ ions in particular) [32] might enhance the interaction between DNA and the analyte by preventing the screening of charges. Additionally, for a similar reason, the functionalization solution was not sonicated once it was prepared since the mechanical force applied might dissociate the two compounds. Finally, the washing of the functionalized electrode was made in PBS and not in the washing solution. Indeed, washing conditions that are too harsh (e.g., involving organic solvents or acids) could possibly destroy the imprinted cavities formed during the functionalization [46].

In the case of dopamine, the current amplitude of the oxidation peak increases and reaches 24 μA at 0.3 V after the electrode functionalization and the peak still remains well defined. For ascorbic acid and serotonin, the peaks are much less defined, while their current amplitudes do not exceed 10 μA at 0.3 V. These data highlight the gain in selectivity for dopamine induced by the use of functionalization based on pre-formed gold nanoparticles.

The triple electrode configuration did not provide significant results when functionalized using the same conditions. Indeed, they do have a very small surface compared to commercial electrodes. Nevertheless, according to the preliminary results obtained using the in-house built potentiostat with non-functionalized electrodes (Figures 2 and 3), it should be possible to lower the detection limit that is currently in the range of 1 mM. In addition, the functionalized electrodes were only tested using well-controlled solutions, without contaminants, and at relatively high concentrations. However, given the selectivity of the proposed method, it should be possible to accurately detect dopamine in more complex solutions, such as cerebrospinal fluid, or in the presence of different molecules or experimental conditions. Moreover, the use of aptamers, based on DNA and RNA, could greatly improve the selectivity of our electrodes because their selection is carried out by an evolutionary process of systematic evolution of ligands by exponential enrichment (SELEX), leading to very specific aptamers with low dissociation constants [47]. Despite the length and the cost associated with this selection, this strategy could be used to improve the electrodes described in this manuscript.

## 4. Conclusions

An in-house built potentiostat was developed and then used in combination with triple electrodes to successfully detect dopamine. Among different tested materials, carbon electrodes allowed the best detection of dopamine. Several experiments involving the functionalization of commercial electrodes were also carried out in order to enhance the selectivity and sensitivity towards a specific neurotransmitter. Pre-formed gold nanoparticles were combined with DNA and polymers in the functionalization solution. While the number of cycles and the gold nanoparticle concentration did not drastically change the performance of the detection, the selectivity for dopamine was considerably improved versus ascorbic acid. In the best case, an increase of 92% and a decrease of 66% of current were observed for dopamine and ascorbic acid, respectively. Improvements should now be achieved to reach the detection of nanomolar concentrations of neurotransmitters [48,49] in order to ensure the applicability of this system to physiological solutions.

**Supplementary Materials:** The following are available online at http://www.mdpi.com/2079-6412/9/8/496/s1, Figure S1: Schematic of the experimental setup of the in-house built potentiostat. ADC, analog to digital converter; BUF, buffer; CE, counter electrode; CF, amplifier feedback capacitor; G, amplifier gain; ID, inner diameter; ISENS, sensed current; OD, outer diameter; PI, proportional integrator electronic circuit; RC, low-pass filter resistor; RE, reference electrode; RF, amplifier feedback resistor; RI, amplifier integrator resistor; RSH, pull down resistor; Rshunt, shunt resistor; SP, buffer used to insulate the input from the electronic circuit; SUB, subtractor amplifier; Vsig, actuation/stimulation signal; WE, working electrode, Figure S2: Geometry of the triple electrodes. 1, carbon triple electrode; 2, stainless triple electrode; 3, tungsten 75 triple electrode; 4, tungsten 125 triple electrode; ID, inner diameter; OD, outer diameter, Figure S3: SEM images of the triple electrodes. 1, carbon triple electrode; 2, stainless triple electrode; 3, tungsten 75 triple electrode; 4, tungsten 125 triple electrode, Figure S4: UV-vis

spectra of gold nanoparticles, Figure S5: Transmission electron microscopy image of gold nanoparticles, Figure S6: Voltammograms obtained for 10 mM dopamine using the carbon triple electrode with the in-house built potentiostat. The values of the maximal sensed current are (A) 0.1 μA, (B) 0.5 μA, (C) 1 μA, (D) 5 μA, (E) 10 μA, and (F) 50 μA, Figure S7: Voltammograms obtained for the analysis of (solid black) 1 mM dopamine and (dotted) 1 mM ascorbic acid for a commercial electrode functionalized with 0.01% m/V HAuCl4 (in situ formation of gold nanoparticles), Table S1: Functionalized electrodes for dopamine detection and their limit of detection (LOD).

**Author Contributions:** Conceptualization, Supervision, Project Administration and Funding Acquisition, A.M. and E.B.; Methodology, Investigation and Analysis M.O. and J.M.; Writing and Editing, M.O., A.M., S.D.N. and E.B.

**Funding:** The authors are indebted to the Natural Sciences and Engineering Research Council of Canada (NSERC), the Eye Disease Foundation, the CHU de Québec Foundation, Discovery and CMC Microsystems' Micro-Nano Technologies program, and Microsystems Strategic Alliance of Quebec for their financial support. E.B. is a research scholar from the Fonds de recherche du Québec–Santé (FRQS) in partnership with the Antoine Turmel Foundation. M.O. was the recipient of a master's scholarship from the FRQS and the Canadian Institutes of Health Research (CIHR).

**Acknowledgments:** We thank J.L. Néron and Y. LeChasseur from Doric Lenses Inc. for discussions and technical support with the electrode design and manufacturing.

**Conflicts of Interest:** The authors declare no conflict of interest.

## Abbreviations

| | |
|---|---|
| Three-electrode setup | 2 carbon (C) fiber electrodes with a diameter of 30 μm and a tip length of 200 μm, and 1 commercial silver/silver-chloride (Ag/AgCl) reference electrode |
| Carbon triple electrode | 2 carbon (C) electrodes with a diameter of 30 μm and a tip length of 200 μm, and 1 platinum–iridium (Pt/Ir) alloy electrode with a diameter of 100 μm and a tip length of 200 μm |
| Commercial Ag/AgCl reference electrode | RRPEAGCL silver/silver-chloride (Ag/AgCl) reference electrode |
| Commercial electrode | RRPE1002C screen-printed carbon electrodes (4 mm × 5 mm carbon working active area electrode) |
| Commercial potentiostat | WaveNow™ potentiostat |
| CTAB | Cetyl trimethylammonium bromide |
| CV | Cyclic voltammetry |
| DNA | Deoxyribonucleic acid |
| DLS | Dynamic light scattering |
| ECL | Electroluminescence |
| FSCV | Fast scan cyclic voltammetry |
| LOD | Limit of detection |
| NaOH | Sodium hydroxide |
| *o*-PD | *o*-Phenylenediamine |
| $PEG_{800}$ | Polyethylene glycol methyl ether thiol with a molecular weight of 800 g·mol$^{-1}$ |
| RNA | Ribonucleic acid |
| SELEX | Systematic evolution of ligands by exponential enrichment |
| SEM | Scanning electron microscopy |
| SERS | Surface-enhanced Raman spectroscopy |
| Simple electrode | 1 carbon (C) fiber electrode with a diameter of 30 μm |
| Stainless triple electrode | 2 stainless steel electrodes with a diameter of 50 μm, and 1 platinum–iridium (Pt/Ir) alloy electrode with a diameter of 100 μm |
| TEM | Transmission electron microscopy |
| Tungsten 75 triple electrode | 2 tungsten (W) electrodes with a diameter of 75 μm, and 1 platinum–iridium (Pt/Ir) alloy electrode with a diameter of 100 μm |
| Tungsten 125 triple electrode | 2 tungsten (W) electrodes with a diameter of 125 μm diameter, and 1 platinum–iridium (Pt/Ir) alloy electrode with a diameter of 100 μm |

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
