# Peer review of "Electrochemical Detection of Dopamine Based on Functionalized Electrodes"

_coatings, doi:10.3390/coatings9080496_

Round 1

Reviewer 1 Report

The authors have achieved the electrochemical detection of dopamine in tandem with home-made triple electrodes and an in-house-built potentiostat at a high scan rate mode. The manuscript is well written with sound discussions and a high level of originality with home-made instrumentation. However, I would recommend the major revision of the following points:

1. In the figures 4 C, D. E anf F, the error bars must be visible to the readers. i recommend the authors to show both (+) and (-) sides of the error bars on clear background.

2. There is no calibration plot for the electrochemical detection of dopamine. The authors seem to have used only three concentrations of dopamine (10 mM, 1 mM and 0.1 mM) and this is not conclusive to draw a calibration plot. I would recommend the authors to prepare a calibration plot with at least four data points and clearly describe how they calculated the Limit of Detection (LOD).

3. There are numerous publications about the electrochemical detection of dopamine using functionalized electrodes. I would recommend the authors to prepare a table with the recent publications and compare the analytical performance of their sensor system with the existing technologies.

In view of my comments above, I would recommend a major revision.

Reviewer 2 Report

The paper by Ouelette et al reports a chemical treatment of commercially electrodes and the combined used of specific potentiostats to ameliorate the detection and sensitivity for dopamine using cyclic voltammetry. For this, they functionalized the commercially available electrodes through the deposition of a thin film containing pre-formed gold nanoparticles. They measured the size of the particles and add other treatments. After having selected carbon electrodes which presented higher sensitivity for dopamine, they tested their treatment and compare the outcome with dopamine and ascorbic acid. They ameliorated the signals with dopamine and not with ascorbic acid (which was even lowered). The profile was also in favor of dopamine when serotonin was present. They conclude that "improvements should now be achieved to reach the detection of nanomolar concentrations of neurotransmitters in order to result in the applicability of this system for physiological solutions”.

Technically, the article has some interest except for those researchers who would be interested by serotonin… Anyway, the paper lacks a functional characterization in a complex system. The article is quite didactic, explains correctly the difficulties, manages several parameters and appropriatly reports the data. Thus, it is quite well written. However, in an assay tube, dopamine has never ever been a neurotransmitter. And the problem is that the amelioration reported for dopamine could also occur for unwanted compounds. What about DOPAC? Since the novelty of the article is not extraordinary (see lines 188-190 and 194-196 for the claims), the applicability of the methods should be tested in complex systems, ultimately in vivo.

The introduction could be shortened a little bit.

Reviewer 3 Report

  The authors reported an electrochemical detection method for dopamine by functionalizing commercially available electrodes through the deposition of a thin film containing preformed gold nanoparticles, DNA and  PEG polymer.

I advise on the  publishing of the manuscript without modifications or corrections

Reviewer 4 Report

Measuring neurotransmitter release with the high spatiotemporal resolution is crucial for understanding how the brain processes and stores information. In the past decade, multiple methods for measuring neurotransmitter concentration in vivo have been developed that already resulted in new insights into the brain function. In the manuscript entitled “Electrochemical Detection of Dopamine Based on Functionalized Electrodes” Ouellette et al. report a simple method for modification of commercially available electrodes with gold nanoparticles. This modification results in the increase in the selectivity of dopamine detection vs vitamin C and serotonin, although with limited sensitivity. Unfortunately, the electrodes with introduced modification have poorly been characterized making it unclear how practical is suggested method. While the manuscript is well written with clear logic and structure, I have several major and minor comments that have to be addressed before publication.

1.     The major limitation of the developed electrodes is its pretty poor sensitivity, about ~1 mM. The authors do not discuss possible ways to improve the sensitivity further to reach a biologically relevant range of dopamine concentrations. And if these methods exist why the authors do not try them?

2.     Further characterization of the dynamic range of dopamine detection is needed. The authors tested only 3 concentration in PBS at pH 7.0!!! I would like to see the performance of the electrode in the condition close to physiological, at least in aCSF, at a wider range of concentration to get a clear idea of the dynamic range of the detection.

3.     The characterization of probe selectivity is also rather limited, only two other substances were tested. I would suggest extending the list of the molecules to tests with some common amino acids and vitamins. Also, in vivo multiple molecules are present simultaneously, it would be also important to see how the sensitivity of the electrode changes in presence of other molecules (measuring the different concentration of dopamine at a constant concentration of vitamin C, for instance).

4.     How does the size of Au nanoparticles influences electrode performance?

5.     How robust is the coating? Does multiple injections of the electrode into the tissue damages or remove the coating? This should be tested and presented in the revised version of the manuscript.

Minor comments:

1.  Please cite optical methods for neurotransmitter detection in vivo as alternatives to electrochemical.

2.  Add the scale bar to Figure 1 and Figure 2.

3.  Please describe dimensions of the electrode and discuss its possible application for measurements in vivo.

4.  Please significantly extend the Discussion part of the manuscript to discuss drawbacks of the methods and its possible further improvements.

5. DNA plays a crucial role in the binding of the analyte. In the study, the authors used DNA from salmon testes? Why? Do DNA length and sequences matter? If so, how?

Reviewer 5 Report

Dear authors,

The manuscript is incorporated with well-organized research results and presented well. Please update the manuscript with the comments suggested below to improve the quality of the manuscript.

1.      Please include the quantification of data in the abstract. For example: Please include the actual increase in the oxidation current for dopamine in line # 18 & 19, or mention the percentage increase, etc.,

2.      The abbreviations should be listed at the end of the manuscript after the conclusion, as per the instructions to authors guidelines of the journal. https://www.mdpi.com/journal/coatings/instructions

3.      Figure-1: Please label the images of the potentiostats components (especially since they are in-house built)

4.      The Schematic of the experimental setup needs to be included with details mentioned in each step.

5.      The SEM image of the triple electrodes (showing the dimensions) as mentioned in section 2.2, should be included in the manuscript.

6.      Figure-2 B & C, the scale bar needs to be included.  The images of the electrodes are not clear and also they need to be shown by itself to make the reader understand the geometry of the electrodes (as mentioned in the review comment-5).

7.      Please include the quantified results in the conclusion as suggested in review comment-1.

Round 2

Reviewer 1 Report

The authors have revised their manuscript in accordance with the comments of the reviewer. Thus, I would recommend the acceptance of the revised manuscript.

Reviewer 2 Report

The authors have correctly answerred to my concern, except for the selectivity of the device toward dopamine.

Reviewer 4 Report

In the revised manuscript the authors performed a few additional experiments to address some of the previous comments. The most important comments were left unaddressed due to the time constraint of the revision process and the overall amount of work required. Although the authors were very upfront in the comments that it was very initial study to test some of their ideas and the developed electrode is far away from implementation in vivo. Unfortunately, the authors were not so upfront in the main text of the manuscript about the current state of the electrode. Before this manuscript can be accepted for publication, the conclusion should clearly state that much more work is required for further improvement and characterization of the electrode for biologically relevant in vitro and in vivo applications, since the current conclusion is misleading  "Improvements should now be achieved to reach the detection of nanomolar concentrations of neurotransmitters [48,49] in order to result in the applicability of this system for physiological solutions". From the authors' responses, it is obvious that it not only improving sensitivity that has to be done.